# Intradermal Immunization of SARS-CoV-2 Original Strain Trimeric Spike Protein Associated to CpG and AddaS03 Adjuvants, but Not MPL, Provide Strong Humoral and Cellular Response in Mice

**DOI:** 10.3390/vaccines10081305

**Published:** 2022-08-12

**Authors:** Luan Firmino-Cruz, Júlio Souza dos-Santos, Alessandra Marcia da Fonseca-Martins, Diogo Oliveira-Maciel, Gustavo Guadagnini-Perez, Victor A. Roncaglia-Pereira, Carlos H. Dumard, Francisca H. Guedes-da-Silva, Ana C. Vicente Santos, Renata G. F. Alvim, Tulio M. Lima, Federico F. Marsili, Daniel P. B. Abreu, Bartira Rossi-Bergmann, Andre M. Vale, Alessandra D’Almeida Filardy, Jerson Lima Silva, Andrea Cheble de Oliveira, Andre M. O. Gomes, Herbert Leonel de Matos Guedes

**Affiliations:** 1Institute of Biophysics Carlos Chagas Filho, Federal University of Rio de Janeiro (UFRJ), Rio de Janeiro 21941-902, RJ, Brazil; 2Institute of Microbiology Paulo de Goes, Federal University of Rio de Janeiro (UFRJ), Rio de Janeiro 21941-902, RJ, Brazil; 3Institute of Medical Biochemistry Leopoldo de Meis, Federal University of Rio de Janeiro (UFRJ), Rio de Janeiro 21941-901, RJ, Brazil; 4National Institute of Science and Technology for Structural Biology and Bioimaging, Federal University of Rio de Janeiro (UFRJ), Rio de Janeiro 21941-901, RJ, Brazil; 5Cell Culture Engineering Lab., COPPE, Federal University of Rio de Janeiro (UFRJ), Rio de Janeiro 21941-598, RJ, Brazil; 6Clinical Immunology Laboratory, Oswaldo Cruz Institute, Oswaldo Cruz Foundation (Fiocruz), Rio de Janeiro 21040-900, RJ, Brazil

**Keywords:** Alum, AddaS03, CpG, MPL, adjuvants, intradermal route, intramuscular route, spike protein, vaccine, SARS-CoV-2

## Abstract

Despite the intramuscular route being the most used vaccination strategy against SARS-CoV-2, the intradermal route has been studied around the globe as a strong candidate for immunization against SARS-CoV-2. Adjuvants have shown to be essential vaccine components that are capable of driving robust immune responses and increasing the vaccination efficacy. In this work, our group aimed to develop a vaccination strategy for SARS-CoV-2 using a trimeric spike protein, by testing the best route with formulations containing the adjuvants AddaS03, CpG, MPL, Alum, or a combination of two of them. Our results showed that formulations that were made with AddaS03 or CpG alone or AddaS03 combined with CpG were able to induce high levels of IgG, IgG1, and IgG2a; high titers of neutralizing antibodies against SARS-CoV-2 original strain; and also induced high hypersensitivity during the challenge with Spike protein and a high level of IFN-γ producing CD4^+^ T-cells in mice. Altogether, those data indicate that AddaS03, CpG, or both combined may be used as adjuvants in vaccines for COVID-19.

## 1. Introduction

Vaccination against SARS-CoV-2 has been extensively shown as an effective method of protection against incidence and hospitalization [1,2,3]. The vast majority of approved vaccines are delivered by the intramuscular (i.m.) route [4]. Nonetheless, many vaccine candidates against SARS-CoV-2 using the intradermal route (i.d.) have shown promising results, either on laboratory tests on different animal models [5,6,7,8], or in development with clinical trials [9,10]. The i.d. route has also been reported as a booster route with less adverse effects compared to i.m. [11]. In addition, it has been shown that the administration of a SARS-CoV-2 epitope-based immunogen through the i.d. route is capable of generating a humoral response and neutralizing activity similar to that of i.m., despite showing a higher capacity of inducing IFN-γ and IL-2 producing CD4^+^ or CD8^+^ T-cells [12].

Adjuvants are key components of many vaccine formulations, since they are capable of eliciting higher immunological activity, increasing antibody production, and T-cell response, thus boosting patients’ response after vaccination [13]. Many adjuvants are nowadays used in approved vaccines, such as potassium aluminum sulfate (Alum), monophosphoryl lipid A (MPL), scalene-based MF59, and 5′-C-phosphate-G-3′ (CpG) 1018 [14]. Adjuvant usage in vaccination strategies against SARS-CoV-2 is absolutely important [15,16]. In fact, many adjuvants have already been evaluated with the spike protein (S ptn) and/or the receptor binding domain (RBD) portion of the S ptn regarding vaccination strategies, such as Matrix-M and AddaVax, both based on scalene [17,18], Alum [7,19], Poly(I:C) [7], and CpG [20]. Nonetheless, ours and other groups have already demonstrated that formulation with two different adjuvants may increase immunization efficacy [7,21]. AddaS03 is a nano-emulsification adjuvant, with a formulation that is based on the adjuvant system AS03, that is capable of inducing transient nuclear factor-κB (NF-κB) activation, the production of cytokines, antigen-specific antibodies, and the migration of immune cells [22,23]. CpG is an adjuvant that is based on a specific DNA motif that is found mostly on bacterial DNA, which is recognized by the toll-like receptor 9 (TLR9), further inducing the production of proinflammatory cytokines and a Type 1 T helper (Th1) cell response [24]. The last tested adjuvant, MPL, is a compound that is derived from lipopolysaccharides of Gram-negative bacteria that are capable of activating TLR4, leading to a Th1-type response [25]. All these adjuvants have already been tested by the i.m. route [26,27,28], however we aimed to evaluate their efficacy along with the S ptn regarding an i.d. route vaccination strategy compared to the i.m. route.

## 2. Methods

### 2.1. Mice

Female BALB/c mice that were 6–8 weeks old (n = 3–5 per group on each experiment), were obtained from the breeding facility of UFRJ. All the animals were kept in mini-isolators (Alesco, São Paulo, Brazil) and kept under controlled temperature and light/dark cycles of 12 h/12 h, in addition to receiving filtered water and commercial feed (Nuvilab, Curitiba, Paraná, Brazil). The experiments were carried out in accordance with the Ethics Committee on the Use of Animals of the Health Sciences Center of the Federal University of Rio de Janeiro (Comitê de Ética no Uso de Animais do Centro de Ciências da Saúde da Universidade Federal do Rio de Janeiro), under the protocol number: 074/20.

### 2.2. Recombinant SARS-CoV-2 Spike Glycoprotein Used as Immunogen

The immunogen that was used is the whole soluble ectodomain (aminoacids 1-1208) of the spike protein (S ptn) of SARS-CoV-2, containing mutations that stabilize it as a trimer in the prefusion conformation. The recombinant HEK293-derived affinity-purified S protein was obtained from the Cell Culture Engineering Laboratory of COPPE/UFRJ. This protein was used in this work to immunize mice and as the ELISA antigen to detect anti-SARS-CoV-2 antibodies in samples from immunized animals.

### 2.3. Immunization

The mice were anesthetized by inhalation of isoflurane (Cristália, Fortaleza, Ceará, Brazil). We immunized mice with a trimeric spike protein (S ptn; 10 μg/dose) associated with adjuvants by the intramuscular route (i.m.) in the muscular area of the right hind footpad or by the intradermal (i.d.) route in the right hind footpad. For that purpose, we used a 1 mL syringe with a 29G needle (BD, Franklin Lakes, NJ, USA). After two weeks, the mice received a booster dose at the same dosage. The control mice received PBS. The formulation that was administrated was as follows: 5 μg of S ptn (5 μL), the nano-emulsion AddaS03^®^ (10 μL), 5 μg of CpG 2395 (5 μL), 5 μg of MPL (5 μL) and 100 μg of Alum (10 μL) per dose. Moreover, homemade sterile phosphate-buffered saline (PBS) was used to make up the final volume of 20 μL when necessary. The S Ptn i.m. (n = 5), S Ptn i.d. (n = 5), S ptn + CPG 2395 i.m. (n = 5), S ptn + MPL i.m. (n = 5), S ptn + AddaS03 i.m. (n = 5), S ptn + MPL + Alum i.m. (n = 5), S ptn + CPG 2395 + Alum i.m. (n = 5), S ptn + CPG 2395 + AddaS03 i.m. (n = 5), S ptn + CPG 2395 i.d. (n = 5), S ptn + MPL i.d. (n = 5), S ptn + MPL + AddaS03 i.d. (n = 5), S ptn + MPL + Alum i.d. (n = 5), S ptn + CPG 2395 + Alum i.d. (n = 5) and S ptn + CPG 2395 + AddaS03 i.d. (n = 5) groups were assessed in one experiment. The PBS i.m. (n = 8) and S ptn + AddaS03 i.d. (n = 10) groups were assessed in two independent experiments. The PBS i.d. (n = 10) group was assessed in three independent experiments.

An experimental timeline is provided in Appendix A.

### 2.4. Antigen-Specific Antibody Responses

Blood was collected from the retro-orbital sinus of immunized mice seven days after prime and each boost. Antigen-specific IgG, IgG1, and IgG2a levels were determined by an enzyme-linked immunosorbent assay (ELISA) using recombinant SARS-CoV-2 Spike protein (S ptn) as a capture antigen. ELISA plates (Corning) were coated with 4 μg/mL of S ptn (in PBS) overnight at 4 °C. The next morning, we discarded the coating solution and added our blocking solution which contained PBS + 5% milk (Molico) for 1 h. Meanwhile, we diluted the serum samples in blocking solution. Then we discarded the blocking solution from the ELISA plates and added the diluted samples for at least 2 h. After this incubation, we washed our ELISA plates 5 times with washing solution which consists of PBS + 0.5% Tween 20 and then added the anti-mouse IgG, IgG1, and IgG2a-HRP (Sigma-Aldrich) detection antibodies for another hour. Then we washed the plate 7 more times and added 3,3′, 5,5’-tetramethylbenzidine (TMB; Invitrogen) for revealing. We stopped the reaction with HCl 1N. The dilutions that were used in IgG assay were 1:3000, 1:9000, 1:27,000, 1:81,000, 1:243,000, 1:729,000, 1:2,187,000, and 1:6,561,000. The dilutions that were used in IgG1 and IgG2a assays were 1:120, 1:360, and 1:1080. The normalized optical density (O.D.) was made by normalizing data from 3 different experiments using the control groups for that end. The O.D. summatory (Sum) was made by summing the values from normalized O.D.

### 2.5. Neutralization Assay

African green monkey kidney cells (Vero, subtype E6) were cultured in high glucose DMEM with 10% fetal bovine serum (FBS; HyClone, Logan, UT, USA), 100 U/mL penicillin, and 100 μg/mL streptomycin (Pen/Strep; ThermoFisher, Walthan, MA, USA) at 37 °C in a humidified atmosphere with 5% CO_2_.

To assess the neutralization titer, the serum samples were incubated with 100 PFU of SARS-CoV-2 with serial dilutions of mouse serum for 1 h at 37 °C (to inactivate mouse serum the samples were heated for 30 min at 56 °C). Then, the samples were placed into 96-well plates with monolayers of Vero cells (2 × 10^4^ cell/well) with supernatants for 1 h at 37 °C. The cells were washed and fresh medium with 2% FBS and 2.4% carboxymethylcellulose was added. After 72 h of infection, the monolayer was fixed with formalin 5% and stained with crystal violet dye solution. The cytopathic effect was scored by independent readers. The reader was blind with respect to the source of the supernatant.

### 2.6. Challenge and Measurement of Hypersensitivity 

After immunization, the BALB/c mice, 6–8 weeks old (n = 5 per group), were challenged subcutaneously in the right hind footpad using a syringe (BD, Franklin Lakes, NJ, USA) with 10 µg of spike protein antigens (S ptn) in 10 µL (1 mg/mL). After the challenge, the hypersensitivity assay was performed, so that the footpad size was also assessed 16 h, 24 h, and 48 h after challenge.

### 2.7. Cell Staining for Flow Cytometry

Cells from the lymph nodes (1 × 10^6^) were stimulated for 4 h at 37 °C with PMA (phorbol 12- myristate 13-acetate, 10 ng/mL, Sigma-Aldrich, Darmstadt, He, Germany) and Ionomycin (10 ng/mL, Sigma-Aldrich, Darmstadt, He, Germany), in the presence of a Golgi complex inhibitor Brefeldin A (5 mg/mL, Biolegend, SanDiego, CA, USA). All centrifugation steps were performed at 4 °C. The cells were washed with PBS and blocked with 50 μL/well Human FcX (BioLegend, SanDiego, CA, USA) for 15 min, after which, 50 μL/well of the staining antibody pool was added and incubated for 30 min at 4 °C. The cells were washed with buffer solution (PBS with 5% FBS) at 400 g for 5 min, then fixed and permeabilized (FoxP3 permeabilization/fixation kit; eBiosicence, Carlsbad, CA, USA) according to the manufacturer’s protocol. The cells were washed again with buffer solution and resuspended in the same solution. For intracellular staining, the following antibodies were added (all used at 0.1 μg/mL) and incubated for 1 h at 4 °C in the dark: CD3 (anti-CD3-BV421, Biolegend, SanDiego, CA, USA), CD4 (anti-CD4-APC-Cy7, Biolegend, SanDiego, CA, USA), CD8 (anti-CD8-Pe-cy7, Biolegend), and intracellular IFN-γ (anti-IFN-γ-APC, eBiosciences, Carlsbad, CA, USA). The cells were washed and resuspended in 150 μL buffer solution and stored in the dark at 4 °C until acquisition. The cells were analyzed on a BD Fortessa™ flow cytometer. The gate strategy was performed based on the selection of cell size (FSC) and composition (SSC). After identifying the main population, a gate of FSC-A (area) and FSC-H (weight) was used, where cellular doublets were excluded. Gates for positive events were established through Fluorescence Minus One (FMO) control. The data analyses were performed using the FlowJo software.

## 3. Results

### 3.1. Immunization Based on Either CpG or AddaS03 (or Both), but Not MPL, Induce Stronger Anti-S ptn IgG Response Thancontrol

In order to understand whether the vaccine formulations were generating an effective B-cell response, we evaluated the serum levels of different anti-S Ptn antibody isotypes. The i.d. route was more effective in inducing S ptn-specific IgG, with most groups exhibiting higher levels than S Ptn, since the first immunization (Figure 1A). Individually, CpG 2395 and AddaS03 showed a strong response by the i.d. route after the first immunization (Figure 1A). In general, combinations containing both AddaS03 and CpG 2395 were able to induce a good response when they were administered by the i.d. route after only one immunization (Figure 1A).

After the second immunization, all the vaccinated groups were able to induce S ptn-specific IgG (Appendix A), except for the S ptn + MPL group by the i.m. route (Figure 1B). In addition, mice that were immunized with either S ptn + CpG 2395 or S ptn + CpG 2395 + AddaS03 (Appendix A) showed stronger IgG production by the i.d. route compared to the i.m. (Figure 1B). 

S ptn + CpG 2395 + AddaS03 group maintained higher levels of IgG by the i.d. route compared to the i.m. route after three immunizations, whereas the S ptn + CpG 2395 exhibited similar profile between both routes (Figure 1B and Appendix A). The group that was immunized with S ptn + AddaS03 also exhibited higher IgG levels by the i.d. route when compared to the i.m. route after the second boost (Figure 1C). Furthermore, once again the group that was immunized with S ptn + MPL failed to induce IgG production by the i.m. route, even though the combination with Alum exhibited higher IgG production by this route (Figure 1C and Appendix A).

### 3.2. Spike Associated to CpG or AddaS03 Induced a Mixed IgG1 and IgG2a and MPL Preferably Induced Igg1 Response

Most formulations induced S ptn-specific IgG1 production after the first boost, with the exception of the S ptn + MPL that was administered by the i.d. route (Figure 2A and Appendix A). After the second boost, all formulations induced IgG1 production (Figure 2B and Appendix A). After two immunizations, the adjuvant CpG 2395 alone or combined with AddaS03 induced higher levels of IgG1 by the i.m. route (Figure 2A and Appendix A), while the AddaS03 alone, MPL + AddaS03, and MPL + Alum were shown to be more effective by the i.d. route (Figure 2A). After the second boost, the differences between the administration routes were maintained only in the S ptn + MPL + AddaS03 formulation by the i.d. route (Figure 2B). Interestingly, immunization with S Ptn alone induced increased IgG1 production after the second boost (Figure 2B).

All the formulations induced S ptn-specific IgG2a by the i.d. route from the first boost, while only S ptn + MPL and S ptn + MPL + Alum were unable to induce IgG2a when they were administered by the i.m. route (Figure 2C and Appendix A). The same was observed after the second boost (Figure 2D and Appendix A). These data suggested that both i.d. and i.m. routes are effective in inducing a humoral response to S ptn and that formulations containing MPL had lower capacity of inducing antibody production.

### 3.3. Immunization by i.d. Route Is More Able to Induce Titers of Neutralizing Antibodies than i.m. Route

We evaluated the in vitro neutralizing activity against SARS-CoV-2 of mice sera that was collected one week after two and three immunizations containing S ptn combined with different adjuvant formulations (Figure 3). First (Figure 3A, left side), we observed that the mice that were immunized twice by the i.m. route with S ptn combined with adjuvant AddaS03 (PRNT50 titer of 281.6) were capable of inducing neutralizing antibodies in relation to PBS, S Ptn, S ptn + CPG 2395, and S ptn + MPL-receiving. However, when AddaS03 was added to the formulation (S ptn + MPL + AddaS03), the mice were able to induce neutralizing antibodies (PRNT50 titer of 614.4). The same effect was observed when AddaS03 was added to the formulation containing S ptn + CpG 2395 (S ptn + CpG 2395 + AddaS03), (PRNT50 titer of 3.600). Besides, when ALUM was added to the formulation S ptn + CpG 2395 (S ptn + CpG 2395 + ALUM), the mice were able to induce neutralizing antibodies (PRNT50 titer of 553.6). There is no difference between PBS and S Ptn-receiving mice.

When we evaluated the ability of immunization by using different combinations of adjuvants with S ptn to induce neutralizing antibodies by the i.d. route (Figure 3A, right side), we observed that this route was able to induce neutralizing antibodies in comparison to S Ptn-receiving mice. Particularly, we found that the combination of S ptn with CpG 2395 (PRNT50 titer of 1056), MPL (PRNT50 titer of 1600) and AddaS03 (PRNT50 titer of 2600) was superior to inducing neutralizing antibodies compared to S Ptn-receiving mice. When AddaS03 and Alum was added to the combination of S ptn + MPL, we observed that these formulations (S ptn + MPL + AddaS03, PRNT50 titer of 1400) and (S ptn + MPL + Alum, PRNT50 titer of 744) were also able to induce neutralizing antibodies in comparison to the S Ptn-receiving mice. Moreover, the combination of S ptn with CpG 2395 + AddaS03 was more efficient in inducing neutralizing antibodies (PRNT50 titer of 3600). Meanwhile, we compared the formulations that were administered by the two different routes and we observed that the combination of S ptn with MPL and AddaS03 by the i.d. route was more able to induce neutralizing antibodies than the i.m. route. There was no difference between the PBS and S Ptn-receiving mice.

Then, we observed the effect of administering different combinations of adjuvants with S ptn after the third immunization by the i.m. route (Figure 3C, left side), we analyzed S ptn + AddaS03-receiving mice were highly endowed to induce neutralizing antibodies in comparison to S ptn-receiving mice (PRNT50 titer of 1689.6). When AddaS03 was added to the formulation S ptn + MPL (S ptn + MPL+ AddaS03), it increased the ability of mice to induce neutralizing antibodies (PRNT50 titer of 1843.2). We also noticed that adding Alum to the combination S ptn + MPL (S ptn + MPL + Alum) and S ptn + CpG 2395 (S ptn + CpG 2395 + ALUM) was able to induce neutralizing antibodies (PRNT50 titer of 1209.6) and (PRNT50 titer of 1604.8), respectively, however, in smaller magnitude in comparison to the S Ptn-receiving mice when AddaS03 was added. We observed that the formulation containing S ptn + CpG 2395 + AddaS03 was superior (PRNT50 titer of 4000) than other the combinations in comparison to S Ptn-receiving mice. Besides, when ALUM was added to the formulation S ptn + CpG 2395 (S ptn + CpG 2395 + ALUM), mice were able to induce neutralizing antibodies (PRNT50 titer of 1604.8). There was no difference between PBS and S Ptn-receiving mice.

Furthermore, when analyzing immunization by the i.d. route (Figure 3C, right side), we observed that combination of S ptn with CpG 2395 (PRNT50 titer of 2800) and AddaS03 (PRNT50 titer of 4000) were fully able to induce neutralizing antibodies than the S Ptn-receiving mice, with higher averages than the second immunization. Besides we saw the S ptn + MPL-receiving mice were unable to induce neutralizing antibodies, the adding of AddaS03 (S ptn + MPL + AddaS03, PRNT50 titer of 3200) and in smaller magnitude Alum (S ptn + MPL + Alum, PRNT50 titer of 1302.4) was capable of inducing neutralizing antibodies. We also noted that a combination of S ptn + CpG 2395 + Alum was able to induce neutralizing antibodies (PRNT50 titer of 1451.2), although at lower levels when compared to the combination S ptn + CpG 2395. In addition, when we added AddaS03 in combination with S ptn + CpG 2395 (S ptn + CpG 2395 + AddaS03), we observed that this formulation was fully able to induce neutralizing antibodies than other vaccine preparations (PRNT50 titer of 4000). Finally, we compared the formulations that were administered by both i.d. and i.m. routes and observed that the combination of S ptn + MPL + AddaS03 and S Ptn + AddaS03 by the i.d. route was more able to induce neutralizing antibodies than the i.m. route. There was no difference between the PBS and S Ptn-receiving mice.

Finally, when we evaluated the in vitro neutralizing activity with PRNT90 by the i.m. and i.d. route one week after two and three immunizations, we observed the similar results with the combinations of adjuvants and demonstrating that S Ptn + CpG 2395 + AddaS03 was more able to induce neutralizing antibodies. There was no difference between the PBS and S Ptn-receiving mice.

### 3.4. S ptn Vaccine Induces a Strong Delayed Hypersensitivity Response When Associated with the Adjuvants AddaS03 + CpG 2395 

To investigate the effect of immunization using S ptn that is associated with the adjuvants, AddaS03 + CpG 2395, since high titers of neutralization was observed, in the delayed-type hypersensitivity (DTH) response in BALB/c mice, S protein was used as challenge and the cellular response was assessed by pachymetry in 16, 24, and 48 h after challenge in groups that were immunized with S ptn that was associated with CpG and AddS03 (Figure 4). In an attempt to increase the immunogenicity of the S ptn vaccine and find the best vaccine candidate, the adjuvants AddaS03 and CpG 2395 were added. The combinations S ptn in association with CpG 2395 and the formulation S ptn and Alum + CpG that were administered by the intramuscular route were not able to induce hypersensitivity. The S ptn formulations in association with the adjuvants AddaS03, CpG, and Alum + CpG that were administered intradermally and the formulation in association with AddaS03 + CpG by the intramuscular route were able to induce a response in the periods of 16 h and 24 h after the challenge, however in 48 h the thickness was reduced. Nevertheless, the association of intradermal S ptn vaccine with AddaS03 + CpG 2395 was capable of inducing a stronger DTH response than all the other groups, maintaining the high response in 48 h, showing that this formulation induces a robust response. 

### 3.5. Evaluation of IFN-γ Production by T-Cells by Vaccine Candidates

We analyzed the immune response that was generated by vaccine candidates from the lymph nodes draining the immunization and challenge. We observed that all the vaccines, with the exception of S ptn + CpG 2395 by the i.d. route, showed a reduction in the frequency of CD4^+^ T-cells when compared to the naive animals (Figure 4B). However, when we observed the number of CD4^+^ T-cells, we saw that the S ptn + CpG 2395 route i.m. and the S ptn + CpG 2395 + AddaS03 route i.d. showed an increase in these cells when compared to the naive animals and that the group that received S ptn + CpG 2395 + AddaS03 route i.d., had a greater number of cells when compared to the other groups that were analyzed (Figure 4C).

We also analyzed the presence of Th1 cells by the production of IFN-γ. The group that received S ptn + CpG 2395 + AddaS03 by the i.m. route and the groups S ptn + AddaS03 and S ptn + CpG 2395 + AddaS03 that were administered by the i.d. route showed a higher percentage of these cells than the naive animals. We also emphasize that the S ptn + CpG 2395 + AddaS03 group by the i.m. route had the highest frequency of these cells among all the other groups that received a vaccine by the same route of administration (Figure 4D). Regarding the number, we saw that the group that S ptn + CpG 2395 + Alum by the i.m. route and the S ptn + CpG 2395 + AddaS03 by the i.d. route had a higher production of IFN-γ by CD4^+^ T-cells (Figure 4E). 

We also checked the amount of CD8^+^ T-cells, known for their importance in cytotoxicity. Our data showed that the S ptn + CpG 2395 + AddaS03 i.m., the S ptn + AddaS03 i.d., and S ptn + CpG 2395 + AddaS03 i.d. groups showed a reduction in the frequency of CD8^+^ T-cells when compared to the I group (Figure 4F). On the other hand, only the S ptn + CpG 2395 i.m., S ptn + CpG 2395 + AddaS03 i.m., and S ptn + CpG 2395 + Alum i.d. groups did not increase the number of these cells after vaccination (Figure 4G). We also evaluated the production of IFN-γ by CD8^+^ T-cells, however there was no difference regarding the frequency and number of these cells in relation to the naive group (Figure 4H,I). 

## 4. Discussion

All the formulations that we tested were able to induce higher levels of serum antibodies than PBS, with the exception of MPL and MPL + Alum (Figure 1 and Figure 2). CpG and AddaS03 were able to induce high levels of antibody in all the formulations that they were present, especially when they were combined (Figure 1 and Figure 2). It is well known that for SARS-CoV and SARS-CoV-2 infections, antibodies are not always protective; in fact, neutralizing antibodies are required to induce protection. CpG and AddaS03 formulations were really effective in inducing high titers of neutralizing antibodies. Once again, the S ptn + CpG 2395 + AddaS03 exhibited the best outcome no matter which route it was injected. However, S ptn + CpG 2395, S ptn + AddaS03 and S ptn + MPL + AddaS03 deserve honor mentions, since they are especially good at inducing neutralizing antibodies responses when administrated by the i.d. route.

Recently, we demonstrated that the immunization of trimetric spike protein of SARS-CoV-2 that was associated to Poly(I:C) plus Alum was able to induce high neutralizing titers against SARS-CoV-2 in vitro by neutralization assay [7] by the i.d. route. Now, we tested formulations with the combination of different adjuvants by the i.d. and i.m. routes (Figure 3). It was showed that two immunizations by the i.m. route with recombinant monomeric SARS-CoV-2 RBD protein in aged and young mice with Alum + CpG 2395 it was able to induce high titers of neutralizing antibodies, while the combination of RBD with AddaS03 failed [21]. Another type of CpG, 1018 in combination Alum by the i.m. induced more neutralizing antibodies than mice that were immunized with RBD + AddaS03 after two immunizations with RBD [29]. Instead of this, the combination with AddaS03 was still able to induce more neutralizing antibodies than mice that were immunized without adjuvants or only Alum. 

A clinical trial using a trimeric form of S ptn, showed that i.m. immunization adjuvanted by AS03, the same compound as AddaS03, induced higher specific IgG and neutralizing antibody levels than the group that was immunized with the S Ptn that was adjuvanted by CPG 1018 + Alum in adults with 55 years old or more after two and three immunizations [30]. Our results showed that the two immunizations with trimetric spike protein by the i.m. route with AddaS03 are capable of inducing neutralizing antibodies. Besides, the combination of S ptn + CpG 2395 + AddaS03 was better to induce high titers of neutralizing antibodies by the i.m. and i.d. routes. Using the i.d. route, only one adjuvant is required as CpG or AddaS03, however, for the intramuscular route the combination with S ptn + CpG 2395 + AddaS03 is recommended. 

A similar strategy with three i.m. immunizations with RBD combined to Alum + MPL or AddaS03 induce similar titers of neutralizing antibodies against SARS-CoV-2B.1.1.7 and B.1.351 variants [31]. We demonstrated that the immunization of S ptn + Alum + MPL was also able to induce neutralizing antibodies, however, to a lesser extent than the formulations containing AddaS03. 

Furthermore, our results showed the two immunizations by the i.d. route are more capable of inducing titers of neutralizing antibodies than the i.m. route with different combinations of adjuvant, but with preference to AddaS03 and CpG or a combination. Besides, three immunizations by i.d. induce even more. These data agree with our recent findings with immunization using Poly(I:C) + Alum [7], demonstrating that Addas03 and CpG 2395 adjuvant may be great candidates for use in vaccines against SARS-CoV-2 by the i.d. route.

All the formulations containing CpG 2395 and AddS03 were able to induce hypersensitivity by the i.d. route, with a peak lesion 16 h after challenge (Figure 4). Thus, it reinforces that immunized animals are able to recognize the antigen and thus induce clonal expansion and a greater recruitment of immune cells to the challenge site, thus evidencing vaccine vigor. AddaS03 and CpG 2395 adjuvants show activity in cell recruitment for both the suggested immunization routes. However, the combination of adjuvants that were associated with S protein had the best performance when administered by the intradermal route when associated with AddaS03 + CpG 2395 than all the other groups over 24 and 48 h period (Figure 4).

T-cell responses are important in the recovery of patients with COVID-19 [32,33] and provide greater long-term protection against SARS-CoV-2 [34]. Evaluations of the T-cell response with the S ptn + CpG 2395 + AddaS03 vaccine that was administered by the i.d. route demonstrated a greater induction of Th1. In studies using the RBD antigen that is associated with the adjuvant CpG 2395 [21] or the adjuvant AddaS03 [31] robust responses to the Th1 profile were demonstrated by the increase in IFN-γ production in CD4^+^ T-cells and CD8^+^ T-cells in response to the immunizer against SARS-CoV-2. Another study using trimeric RBD + Addavax as immunization showed higher neutralizing antibodies titers after the first, second, and third doses and it was followed by a higher percentage of T CD4^+^IFN-γ^+^ cells in the peripheral blood of macaques [35]. It has also been shown that immunized macaques had less viral load in the rectum, trachea, and throat after the challenge when compared to the non-immunized group [35]. Our candidate S ptn + CpG 2395 + AddaS03 did not show an increase in the IFN-γ response by CD8^+^ T-cells. The participation of Th1 can be demonstrated by the induction of IgG2a when used CpG or AddS03 as adjuvants.

## 5. Conclusions

This study has its limitations regarding the low number of mice that were used, the SARS-CoV-2 strain that was used for the neutralization assays and the lack of infection of immunized mice. However, even with all the limitations, the preliminary data showed that the association of S ptn + CpG 2395 + AddaS03 by the i.d. and i.m. routes demonstrated the ability to induce robust humoral and cell responses, and together with S ptn + CpG 2395 and S ptn + AddaS03 could be good candidates to vaccines against the original strain of SARS-CoV-2 when administrated preferentially by the i.d. route.

## Figures and Tables

**Figure 1 vaccines-10-01305-f001:**
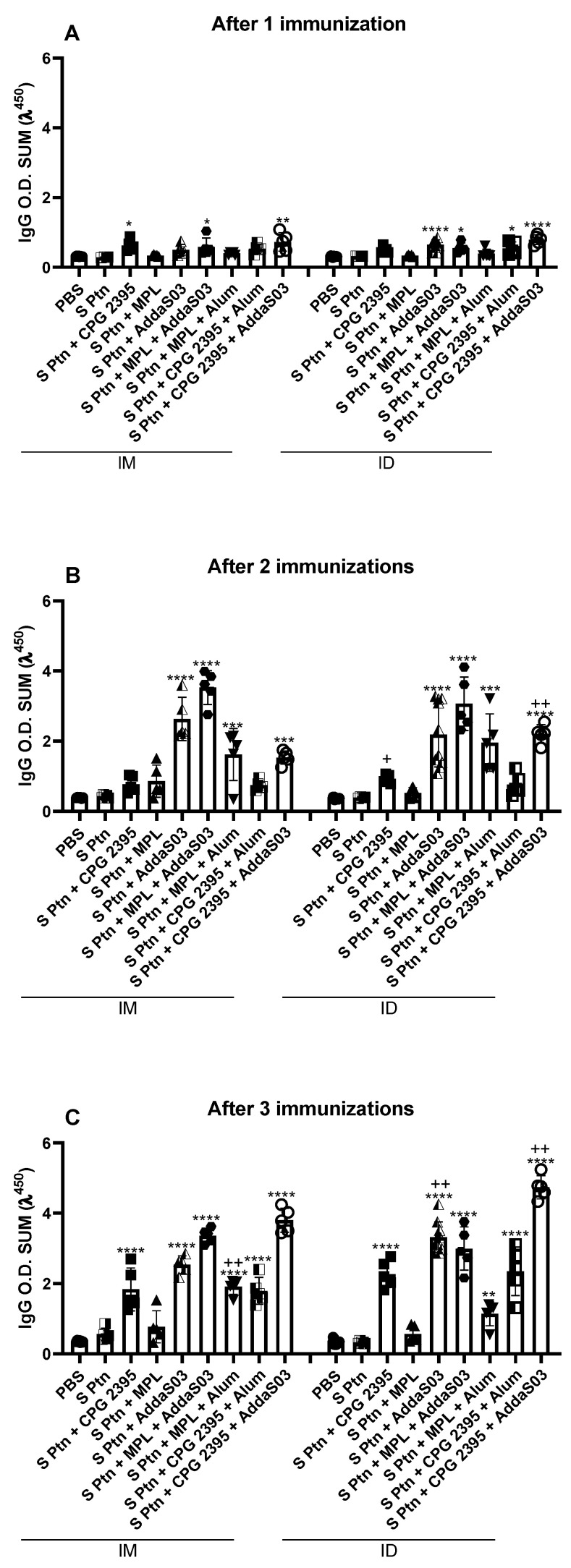
CpG 2395 and AddaS03 based formulations induce strong anti-S Ptn IgG responses. Mice were immunized with a 14-day interval between doses and blood samples were taken 7 days after each immunization. The serum levels of IgG were assessed by ELISA after 1 (**A**), 2 (**B**), and 3 (**C**) immunizations. * Differences between the immunized versus the S Ptn group; + differences between the i.d. and i.m. route. * or + *p* < 0.05; ** or ++ *p* < 0.01; *** *p* < 0.001; **** *p* < 0.0001. One way ANOVA test. Normalized data from three independent experiments are shown. The S Ptn i.m. (n = 5), S Ptn i.d. (n = 5), S ptn + CPG 2395 i.m. (n = 5), S ptn + MPL i.m. (n = 5), S ptn + AddaS03 i.m. (n = 5), S ptn + MPL + Alum i.m. (n = 5), S ptn + CPG 2395 + Alum i.m. (n = 5), S ptn + CPG 2395 + AddaS03 i.m. (n = 5), S ptn + CPG 2395 i.d. (n = 5), S ptn + MPL i.d. (n = 5), S ptn + MPL + AddaS03 i.d. (n = 5), S ptn + MPL + Alum i.d. (n = 5), S ptn + CPG 2395 + Alum i.d. (n = 5), and S ptn + CPG 2395 + AddaS03 i.d. (n = 5) groups were assessed in one experiment. The PBS i.m. (n = 8) and S ptn + AddaS03 i.d. (n = 10) groups were assessed in two independent experiments. The PBS i.d. (n = 10) group was assessed in three independent experiments.

**Figure 2 vaccines-10-01305-f002:**
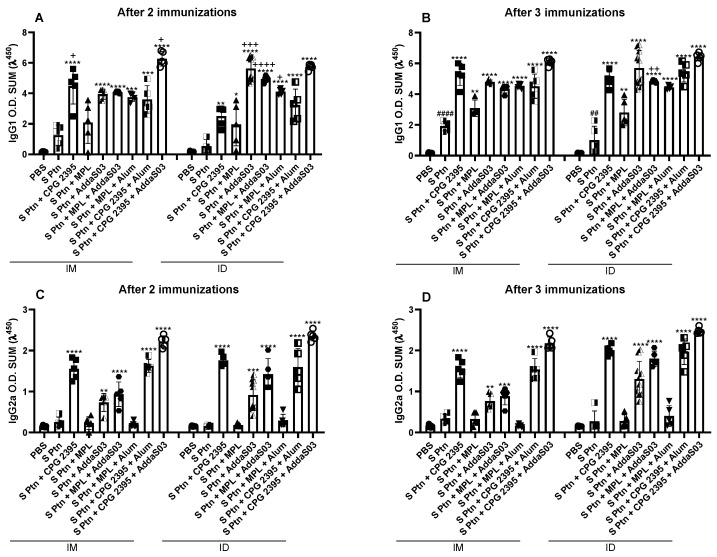
CpG 2395 and AddaS03 based formulations induce higher levels of anti-S Ptn IgG subtypes. Mice were immunized with a 14-day interval between doses and blood samples were taken 7 days after each immunization. The serum levels of IgG1 and IgG2a were assessed by ELISA after 2 ((**A**,**C**) respectively), and 3 ((**B**,**D**) respectively) immunizations. # Differences between S Ptn and PBS group; * differences between the immunized versus the S Ptn group; + differences between the i.d. and i.m. route. * or + *p* < 0.05; **, ## or ++ *p* < 0.01; *** or +++ *p* < 0.001; ****, #### or ++++ *p* < 0.0001. One way ANOVA test. Normalized data from three independent experiments are shown. The S Ptn i.m. (n = 5), S Ptn i.d. (n = 5), S ptn + CPG 2395 i.m. (n = 5), S ptn + MPL i.m. (n = 5), S ptn + AddaS03 i.m. (n = 5), S ptn + MPL + Alum i.m. (n = 5), S ptn + CPG 2395 + Alum i.m. (n = 5), S ptn + CPG 2395 + AddaS03 i.m. (n = 5), S ptn + CPG 2395 i.d. (n = 5), S ptn + MPL i.d. (n = 5), S ptn + MPL + AddaS03 i.d. (n = 5), S ptn + MPL + Alum i.d. (n = 5), S ptn + CPG 2395 + Alum i.d. (n = 5), and S ptn + CPG 2395 + AddaS03 i.d. (n = 5) groups were assessed in one experiment. The PBS i.m. (n = 8) and S ptn + AddaS03 i.d. (n = 10) groups were assessed in two independent experiments. The PBS i.d. (n = 10) group was assessed in three independent experiments.

**Figure 3 vaccines-10-01305-f003:**
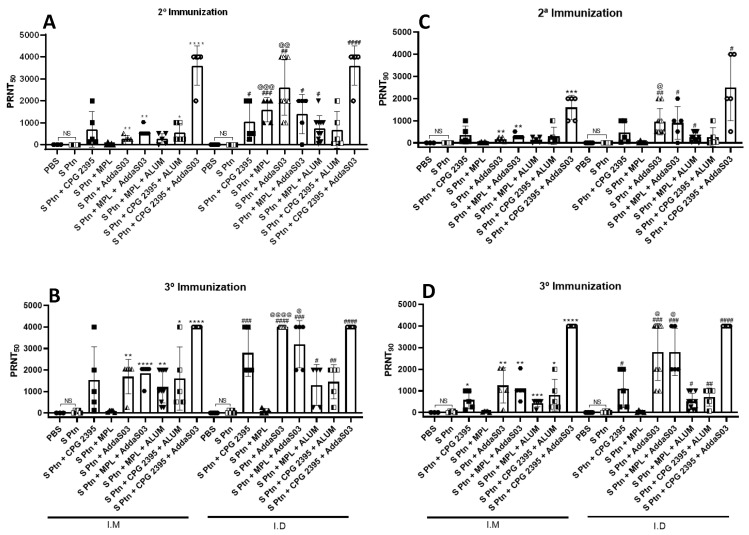
S ptn combined with CpG 2395 + AddaS03 is more capable of inducing titers of neutralizing antibodies. The serum samples were collected as described in the materials and methods after 2° (**A**,**C**) and 3° (**B**,**D**) immunization, (PRNT50) and (PRNT90). The symbol (*) represents the comparison in relation to the S Ptn group by the i.m. route, (#) represents the comparison in relation to the S Ptn group by the i.d. route, and (@) represents the comparison between each group by the i.m. and i.d. route. The data are presented as the mean ± standard error of the mean that were analyzed by unpaired *t*-test, *, # or @ *p* < 0.05, **, ## or @@ *p* < 0.01, ***, ### or @@@ *p* < 0.001, ****, #### or @@@@ *p* < 0.0001. NS; No statistic. Data from two independent experiments are shown. The PBS i.m. (n = 3), S ptn + CPG 2395 i.m. (n = 5), S ptn + MPL i.m. (n = 5), S ptn + AddaS03 i.m. (n = 5), S ptn + MPL + Alum i.m. (n = 5), S ptn + CPG 2395 + Alum i.m. (n = 5), S ptn + CPG 2395 + AddaS03 i.m. (n = 5), S ptn + CPG 2395 i.d. (n = 5), S ptn + MPL i.d. (n = 5), S ptn + MPL + AddaS03 i.d. (n = 5), S ptn + MPL + Alum i.d. (n = 5), S ptn + CPG 2395 + Alum i.d. (n = 5), and S ptn + CPG 2395 + AddaS03 i.d. (n = 5) groups were assessed in one experiment. The PBS i.d. (n = 10) and S ptn + AddaS03 i.d. (n = 10) were assessed in two independent experiments.

**Figure 4 vaccines-10-01305-f004:**
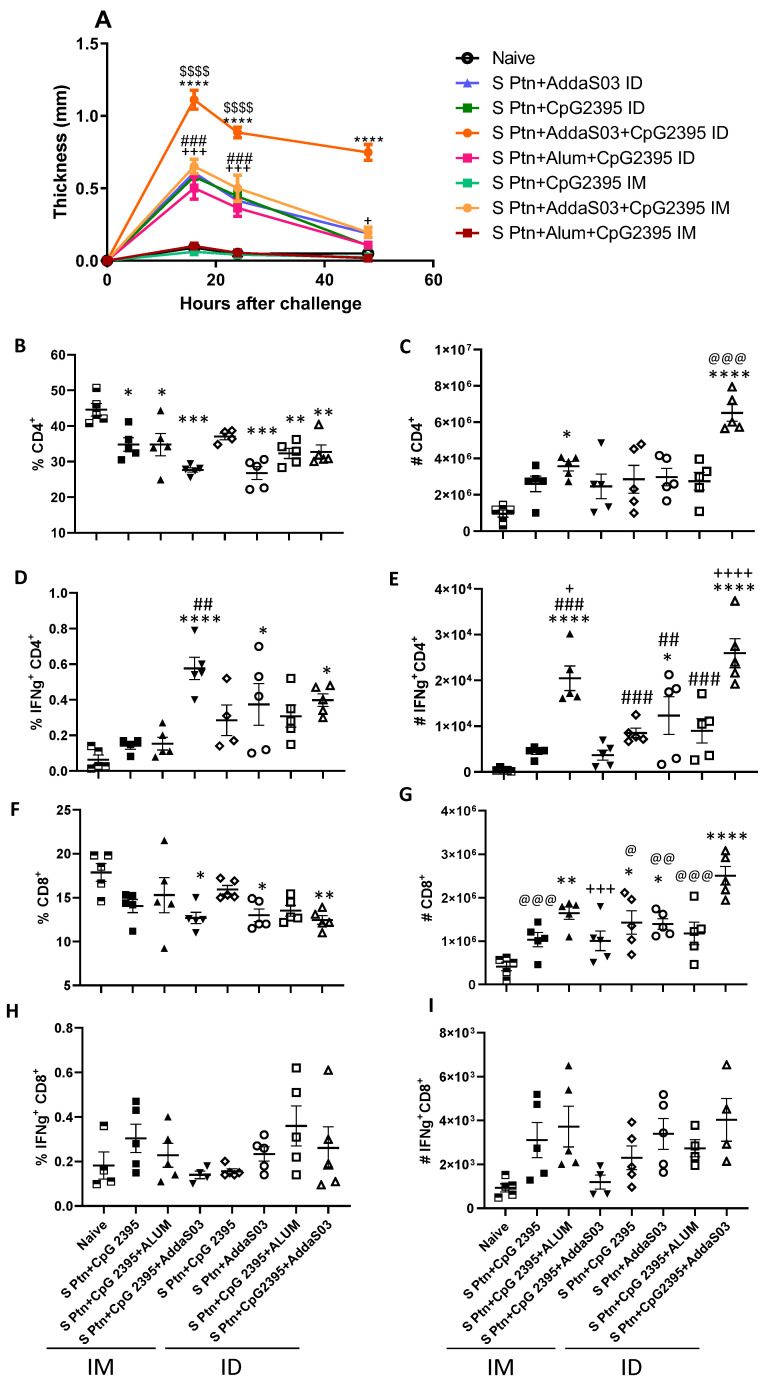
Intradermal S ptn vaccine that was associated with the adjuvants AddaS03 + CpG 2395 induces a strong delayed hypersensitivity response and the production of IFN-γ by CD4^+^ T-cells. The mice were immunized with S ptn with AddaS03 and CpG 2395 alone or together, while the controls received PBS alone. The animals were then challenged with S ptn in the right hind footpad and the kinetics of the hypersensitivity response (**A**) were scored at 16, 24, and 48 h post-infection. Lymphocytes from BALB/c mice were collected from the popliteal draining lymph node of the immunized paw or via the i.m. or i.d. route after x time. Non-immunized mice were used as a control. The mice were immunized by the i.m. route with S ptn + CpG 2395; S ptn + CpG 2395 + Alum; S ptn + CpG 2395 + AddaS03, and by the i.d. route S ptn + CpG 2395; S ptn + AddaS03; S ptn + CpG 2395 + Alum; S ptn + CpG 2395 + AddaS03. The cells were analyzed by flow cytometry. (**B**) Percentage of CD4^+^ T-cells. (**C**) Number of CD4^+^ T-cells. (**D**) Percentage of IFN-γ^+^ CD4^+^ T-cells. (**E**) Number of IFN-γ^+^ CD4^+^. (**F**) Percentage of CD8^+^ T-cells. (**G**) Number of CD8^+^ T-cells. (**H**) Percentage of IFN- γ^+^ CD8^+^ T-cells. (**I**) Number of IFN- γ^+^ CD8^+^ T-cells. The data + SEM of individually mice (5 mice/group). In (**A**): * represents differences between S ptn + CpG 2395 + AddaS03 i.d. and every other group; + represents the differences between naïve and either S ptn + AddaS03 i.d. or S ptn + CpG 2395 + AddaS03; # represents the differences between naïve and either S ptn + CpG i.d. or S ptn + CpG 2395 + Alum i.d.; $ represents the differences between either S ptn + CpG 2395 + Alum i.d. versus S ptn + CpG 2395 + Alum i.m. or S ptn + CpG 2395 i.d. versus S ptn + CpG 2395 i.m.. In (**B**–**I**): * represents the differences between the naïve and vaccinated groups; + represents the differences between mice that were vaccinated with the same formulation by different routes; # differences between the group with higher levels or numbers that were vaccinated by either i.m. or i.d. and the other groups that were vaccinated by the same route; @ differences between S ptn + CpG 2395 + AddaS03 i.d. and all the other groups. The percentage was obtained by gating the CD4^+^ or CD8^+^ populations and the percentage of IFN- γ^+^ cells were gated within the CD4^+^ or CD8^+^ population. * or @ *p* < 0.05, **, ## or @@ *p* < 0.01, ***, ###, +++ or @@@ *p* < 0.001, ****, ++++ or $$$$ *p* < 0.0001. The number of cells were calculated by the equivalency of the percentages in the total number of cells in the dLN of each mouse. The data shown are from one experiment. (n = 5 per group).

## Data Availability

Data are available by contact with correspondent author.

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
