# Peer review of "Intradermal Immunization of SARS-CoV-2 Original Strain Trimeric Spike Protein Associated to CpG and AddaS03 Adjuvants, but Not MPL, Provide Strong Humoral and Cellular Response in Mice"

_vaccines, 2022, doi:10.3390/vaccines10081305_

Round 1
Reviewer 1 Report
The authors investigated the ability in generating antibody and cellular response of intradermal route delivery of anti SARS CoV-2 vaccines .
They tested different adjuvants combination concluding that intradermal route and the addition of AddaS03 and CpG2395 to S ptn is able to increase anti SARS antibody and cellular response.
The paper is very interesting, experiments well though, results and figure are clear, and reference updated .
I have just few questions:
1. Intradermal route seemed to induce high neutralizing IgG levels. How about IgG persistence compared to the conventional IM route?
2. The Id route with AddaS03 and CpG2395 generated a strong delayed hypersensitivity reaction, suggesting a robust response. Is this compatible with human administration?. What about other possible side effects due to adjuvants?
3. Evaluation of T cell subpopulations suggest high IFN-g production, confirming also a cell mediated immune response. Did they have any idea in the generation of memory T and B cells?
Author Response
Comments and Suggestions for Authors (Reviewer 1)
The authors investigated the ability in generating antibody and cellular response of intradermal route delivery of anti SARS CoV-2 vaccines .
They tested different adjuvants combination concluding that intradermal route and the addition of AddaS03 and CpG2395 to S ptn is able to increase anti SARS antibody and cellular response.
The paper is very interesting, experiments well though, results and figure are clear, and reference updated.
Authors: Thank you very much for your considerations, we really appreciate it.
I have just few questions:
- Intradermal route seemed to induce high neutralizing IgG levels. How about IgG persistence compared to the conventional IM route?
Authors: That is a very good question. Usually, immunization using protein associated to adjuvant with 3 doses induce a last long immunity by i.m route, and should be similar in i.d. immunization. We haven’t assessed it because those data are preliminary (communication paper). This is something we definitively should check it in a future paper. Thank you for the insight.
- The Id route with AddaS03 and CpG2395 generated a strong delayed hypersensitivity reaction, suggesting a robust response. Is this compatible with human administration? What about other possible side effects due to adjuvants?
Authors: Thank you for the questions. We don’t know how this combination act in humans, but we do know that all formulations containing AddaS03 induced inflammation on the site of administration. Studies to comprehend the safety of the use AddaS03 by i.d. route should be performed in future pre-clinical studies.
- Evaluation of T cell subpopulations suggest high IFN-g production, confirming also a cell mediated immune response. Did they have any idea in the generation of memory T and B cells?
Authors: Thank you for this question. As said before, those are preliminary data ang by our point of view, some formulations present highly promising characteristics for further studies. This time, we did not invest time to check those populations, but it should be done in the future in a paper containing more definitive data.

Reviewer 2 Report
I have read with great interest the article entitled "Intradermal immunization of SARS-CoV-2 original strain trimeric spike protein associated to CpG and AddaS03 adjuvants, but not MPL, provide strong humoral and cellular response in mice"
In the manuscript, the authors perform experiments with a murine experimental model using a common strain of allergic disease due to its characteristic potent humoral immune response. The experiments are well designed and although the number of samples is certainly low, their results are promising.
In the experiments I miss having a control group with the use of only one adjuvant; Having a control group with the use of CPG alone and MPL alone would have really shown if both adjuvants are important by themselves for the different results obtained. In other experimental models if there are differences.de la Torre MV, Baeza ML, Nájera L, Zubeldia JM. Comparative study of adjuvants for allergen-specific immunotherapy in a murine model. Immunotherapy. 2018 Oct;10(14):1219-1228. doi: 10.2217/imt-2018-0072. Epub 2018 Sep 24. PMID: 30244623.
Another improvement for future experiments would be the use of other strains with a greater load of non-humoral response or with a tendency to more subtle differences, such as the use of a B6 strain or even beyond strains such as B10.RIII used for models of autoimmune diseases. The importance of the strain will give more force to these results. Brewer JP, Kisselgof AB, Martin TR. Genetic variability in pulmonary physiological, cellular, and antibody responses to antigen in mice. Am J Respir Crit Care Med. 1999 Oct;160(4):1150-6. doi: 10.1164/ajrccm.160.4.9806034. PMID: 10508801.
Even for experimental models of hypersensitivity there are different strains that could provide results with subtle differences in order to obtain more information.
Marco-Martín G, La Rotta Hernández A, Vázquez de la Torre M, Higaki Y, Zubeldia JM, Baeza ML. Differences in the Anaphylactic Response between C3H/HeOuJ and BALB/c Mice. Int Arch Allergy Immunol. 2017;173(4):204-212. doi: 10.1159/000478983. Epub 2017 Aug 30. PMID: 28850948.
But in conclusion, it seems to me a good manuscript, with very interesting results and that opens a path for future development.
Author Response
Authors:
Thank you for the considerations, we really appreciate them.
I don’t know if I understood completely what you meant by your comment on control groups. But if you meant the we should have experimental groups with only the adjuvant and without the antigen, our dataset around antibody production is always around virus specific antibodies (anti-spike protein or neutralizing antibodies). For that reason, we did not have experimental groups with only the adjuvants, since it would require virus particles to induce any specific antibodies. That is why we agreed that PBS group would perform the role of negative control very well. I have added some text in green to try to make more clear that we are always checking anti-spike protein antibodies.
If you meant that we should have experimental groups with one adjuvant associated to the antigen, we do have those controls. I ask you to look carefully into figures 1 to 3: the columns 3 to 5 (CPG alone, MPL alone and AddaS03 alone associated to S Ptn administrated intramuscularly respectively) and 12 to 14 (CPG alone, MPL alone and AddaS03 alone associated to S Ptn administrated intradermally respectively) represent those groups.
Since this is a preliminary data paper (communication paper), we did not check everything that we think that is important. We absolutely agree that is necessary to check the responses we observed in BALB/c, in other mice and mammal models to validate these formulations as potential vaccine candidates, but this is something we intend to do in future studies.
Thank you so much for the references that you suggested our read.
Once again, we really appreciate the considerations and we hope to have responded and justified your concerning around it.

This manuscript is a resubmission of an earlier submission. The following is a list of the peer review reports and author responses from that submission.
Round 1
Reviewer 1 Report
The manuscript of Firmino-Cruz et al. aimed at addressing if i.d. injection was better than i.m. and which adjuvant works better during immunization with Spike protein of SARS-CoV-2. They analyzed both humoral (in terms of total IgG and subclasses) and cellular (DTH and IFNg CD4+ and CD8+ T producing cells) immune response in i.d. and i.m immunization with four types of adjuvant or their combinations. However, the major concern of all the experiments is the lacking of a proper controls, first of all immunization with only S Ptn. How authors can conclude that “the association of S Ptn + CpG 2395 + AddaS03 by the ID and IM routes demonstrated the ability to induce a robust humoral and cell responses” by comparing it only with mice receiving PBS and not with mice receiving S Ptn? This is a control absolutely required for addressing the ability of an adjuvant in increase the immune response. I strongly recommend to add results about i.m. and i.d. immunization with S Ptn.
Then, immunization with S Ptn must be introduced in the abstract. The reader does not understand the type of vaccination used until the methods section. Furthermore, which type of Spike protein was used? Produced by the authors? Or manufacturing by? This should be clarified in the method section.
How many mice were immunized per group? How many times the experiment was performed? Numbers should be added both in methods and in the figure legends.
Which kind of mice were used? Please clarify it in the methods.
One week after immunization is a relative short time to evaluate both cellular and humoral immune responses. However, it could be accepted if other evaluations will be performed at a greater distance of time (i.e. 2 months after the third vaccination). This will be helpful in the evaluation of adjuvant ability to induce a long lasting immune response, that is a critical issue to be addressed in this moment of the vaccination campaign.
The cellular immune response elicited by vaccination should be addressed by restimulating T cells with the spike protein. Why authors decided to use PMA plus ionomycin?
Minors:
Description of Vero cells can be included into the description of neutralization assay.
Figure 1: The three graphs can be aligned horizontally in order to allow an easier understanding.
Figure 4: How percentage and numbers were calculated?
Author Response
Letter to the Reviewers
Dear reviewers,
We hope to find you well. Thanks for all comments and suggestions, we tried our best to achieve all of your suggestions and we know that made our work better.
Here we will comment all the suggestions made and how we proceeded to fulfill them.
Answer for reviewer 1:
Comments and Suggestions for Authors
The manuscript of Firmino-Cruz et al. aimed at addressing if i.d. injection was better than i.m. and which adjuvant works better during immunization with Spike protein of SARS-CoV-2. They analyzed both humoral (in terms of total IgG and subclasses) and cellular (DTH and IFNg CD4+ and CD8+ T producing cells) immune response in i.d. and i.m immunization with four types of adjuvant or their combinations.
Dear reviewer, we really appreciate your comments and opinions on our work. Thank you very much for all of them.
However, the major concern of all the experiments is the lacking of a proper controls, first of all immunization with only S Ptn. How authors can conclude that “the association of S Ptn + CpG 2395 + AddaS03 by the ID and IM routes demonstrated the ability to induce a robust humoral and cell responses” by comparing it only with mice receiving PBS and not with mice receiving S Ptn? This is a control absolutely required for addressing the ability of an adjuvant in increase the immune response. I strongly recommend to add results about i.m. and i.d. immunization with S Ptn.
We understand your concerning but we tested these controls on a previous pre-print work (Souza dos-Santos, 2021), which is also being reviewed for publication. The S ptn group had very few differences comparing to the PBS group (sometimes no difference at all) in the ID immunization. We agree that this would be the best control in this case, but considering that we decided not to do that this time.
Souza dos-Santos, J., Firmino-Cruz, L., Marcia da Fonseca-Martins, A., Oliveira-Maciel, D., Guadagini Perez, G., R Pereira, V. A., Dumard, C. H., Guedes-da-Silva, F. H., Vicente Santos, A. C., Souza dos-Santos Leandro, M., Rafael Machado Ferreira, J., Guimarães-Pinto, K., Conde, L., S Rodrigues, D. A., Vinicius de Mattos Silva, M., F Alvim, R. G., Lima, T. M., Marsili, F. F., B Abreu, D. P., … Leonel de Matos Guedes, H. (2021). Immunogenicity of trimetric spike protein associated to Poly(I:C) plus Alum. BioRxiv : The Preprint Server for Biology. https://doi.org/10.1101/2021.10.05.461434
Then, immunization with S Ptn must be introduced in the abstract. The reader does not understand the type of vaccination used until the methods section. Furthermore, which type of Spike protein was used? Produced by the authors? Or manufacturing by? This should be clarified in the method section.
We added that information in the abstract and methodology section as asked. We really missed that information.
How many mice were immunized per group? How many times the experiment was performed? Numbers should be added both in methods and in the figure legends.
We have now added this information in both methodology section and figure legends.
Which kind of mice were used? Please clarify it in the methods.
We used female BALB/c mice and we have added this information in the methods section, in fact we added a new subject to the methods section containing this information.
One week after immunization is a relative short time to evaluate both cellular and humoral immune responses. However, it could be accepted if other evaluations will be performed at a greater distance of time (i.e. 2 months after the third vaccination). This will be helpful in the evaluation of adjuvant ability to induce a long lasting immune response, that is a critical issue to be addressed in this moment of the vaccination campaign.
We agree that this would be very interesting to study memory, but since this is a preliminary study, we ask you to reconsider. It is going to be a future study.
The cellular immune response elicited by vaccination should be addressed by restimulating T cells with the spike protein. Why authors decided to use PMA plus ionomycin?
We used PMA + Ionomycin because it is a common unspecific stimulation method used to this kind of analysis. We tried to use spike, unfortunately, did not work. We understand that using spike protein would be better as a specific inducer, but PMA + ionomycin in broadly used and we think it would be enough to evaluate our immunization setup. Other option could be peptides from spike, but we did not have during our experiments.
Minors:
Description of Vero cells can be included into the description of neutralization assay.
We have done it as you asked and we think it is better this way.
Figure 1: The three graphs can be aligned horizontally in order to allow an easier understanding.
We tried it and it made the graphs look really small, so we went back to the original one.
Figure 4: How percentage and numbers were calculated?
Percentages were calculated in-gate, meaning the percentage of IFN+ was calculated either within a CD4+ or CD8+ cells pre-gate, which were calculated within a single cell pre-gate, which was calculated within a cell of interest pre-gate.
The number of cells were calculated by multiplying the total number of cells of each lymph node to the percentage of each pre-gate until the percentage of the gate we showed in the graph.
Those information were added to the figure legend.

Reviewer 2 Report
Vaccines – Manuscript ID: 1647126 (Communication)
Thank you very much for the opportunity given to me to review the manuscript entitled “Intradermal immunization of SARS-CoV-2 trimetric spike protein associated to CPG and AddaS03, but not MPL, provide strong humoral and cellular response” by Firmino-Cruz et al. that I found extremely interesting I found it extremely interesting with respect to the efficacy of COVID-19 vaccination by intradermal administration. Importantly, this study showed a difference in this efficacy depending on the composition of the adjuvants.
However, there are some points that need revision:
TITLE
- The authors are invited to add the word “adjuvants” to their title.
- The experiments were performed on mice not human-being, and this information have to appear in the TITLE and the ABSTRACT sections.
INTRODUCTION
- The name of the adjuvants should be written with the same manner in all the manuscript (for example, CpG or CPG) including keywords (ADDS03, while AddaS03 in the title and the abstract), and mention them in the same order. In the title for example CpG was before AddaS03m, but not on line 35.
- Lines 35-36: it is not clear if you have combine CPG with CPG. Please, verify and/or correct?
- Line 62: The mentioned reference “CDC, 2020” should be listed in REFERENCES section. Same comment for “Tian JH et al., 2020”, “Amanat F et al., 2020 »... and all the cited references. The authors are also invited to verify, for each cited reference, the concordance between the manuscript and the REFERENCES List (for example: reference 22 was cited as “dos Santos et al., 2021” sometimes in the manuscript, and “Souza dos-Santos, J. 2021” in the list.
- The names of all adjuvants:
- Should be detailed, as it was done with MPL (line 62),
- And written similarly when they were cited. For example, Monophosphoryl lipid A (MPL), and Monophosphoryl-Lipid A (MPLA).
- Line 65: RBD abbreviation should me detailed when first mentioned in the manuscript. Same comment for other abbreviations, which could be written always with the same manner (for example: ptn/Ptn).
MATERIALS AND METHODS
- Line 85: The number of mice was not mentioned.
- Line 90: PBS word should be detailed, and the manufacturer references should be mentioned. Same comment for TMB (line 111), FBS (line 123).
- Line 91:
- The quantity used (µg) of AddaS03 adjuvant was not mentioned.
- The concentration of the used adjuvants was not the same, why? This point is critical as it can have an influence on the efficacy of the tested adjuvants.
- MPL is Alum (line91)? It was used
- Line 120: On which variants the study was performed? This information should also be stated on the title and in the abstract.
- Line 122: Is the number 4 of “2x104 cell/well” exponent?
- Lines 154 and 155:
- The authors used the adjective “stronger”, but they do not mention comparing to what.
- This sub-title is not clear: was CPG alone, AddaS03 alone or the association of both of them was stringer? Please, precise.
RESULTS
- Line 274: The authors stated “… 18, 24 and 48 h after challenge…”, however it was 16, 24 and 48h in the Supplementary Figure 1. This information should be corrected.
DISCUSSION
- Well written. However, the limitations of this study were not listed. These results were not obtained with different variants of SARS-CoV-2, tested on a large number of mice, the tested mice were not really infected, but only a small part of its spike was used as epitopic, therefore the results cannot represent what happens in infected mice…
Author Response
Letter to the Reviewers
Dear reviewers,
We hope to find you well. Thanks for all comments and suggestions, we tried our best to achieve all of your suggestions and we know that made our work better.
Here we will comment all the suggestions made and how we proceeded to fulfill them.
Answer for Reviewer 2
Comments and Suggestions for Authors
Vaccines – Manuscript ID: 1647126 (Communication)
Thank you very much for the opportunity given to me to review the manuscript entitled “Intradermal immunization of SARS-CoV-2 trimetric spike protein associated to CPG and AddaS03, but not MPL, provide strong humoral and cellular response” by Firmino-Cruz et al. that I found extremely interesting I found it extremely interesting with respect to the efficacy of COVID-19 vaccination by intradermal administration. Importantly, this study showed a difference in this efficacy depending on the composition of the adjuvants.
Dear reviewer, we really appreciate your comments and opinions on our work. Thank you very much for all of them. We would also like to thank you for your kind positive feedback about our work.
However, there are some points that need revision:
TITLE
The authors are invited to add the word “adjuvants” to their title.
We added it to the title as suggested.
The experiments were performed on mice not human-being, and this information have to appear in the TITLE and the ABSTRACT sections.
We have also added this information in both title and abstract.
INTRODUCTION
The name of the adjuvants should be written with the same manner in all the manuscript (for example, CpG or CPG) including keywords (ADDS03, while AddaS03 in the title and the abstract), and mention them in the same order. In the title for example CpG was before AddaS03m, but not on line 35.
Thanks for warning us of that. We have done that and we think now it is all standardized along the manuscript.
Lines 35-36: it is not clear if you have combine CPG with CPG. Please, verify and/or correct?
We use CPG and AddaS03 separately or together. Both are formulations that we used.
Line 62: The mentioned reference “CDC, 2020” should be listed in REFERENCES section. Same comment for “Tian JH et al., 2020”, “Amanat F et al., 2020 »... and all the cited references. The authors are also invited to verify, for each cited reference, the concordance between the manuscript and the REFERENCES List (for example: reference 22 was cited as “dos Santos et al., 2021” sometimes in the manuscript, and “Souza dos-Santos, J. 2021” in the list.
We have standardized it and also added the missing references.
The names of all adjuvants:
Should be detailed, as it was done with MPL (line 62), and written similarly when they were cited. For example, Monophosphoryl lipid A (MPL), and Monophosphoryl-Lipid A (MPLA).
We have made the alterations in the manuscript.
Line 65: RBD abbreviation should me detailed when first mentioned in the manuscript. Same comment for other abbreviations, which could be written always with the same manner (for example: ptn/Ptn).
Everything should be set up after the alterations made.
MATERIALS AND METHODS
Line 85: The number of mice was not mentioned.
We missed that but now it is described in the methods and also in the figure legends.
Line 90: PBS word should be detailed, and the manufacturer references should be mentioned. Same comment for TMB (line 111), FBS (line 123).
We have used homemade sterile PBS and added the full names of PBS, TMB and FBS as asked.
MENCIONADO ANTERIORMENTE NA LINHA 111
Line 91:The quantity used (µg) of AddaS03 adjuvant was not mentioned.
It is not possible to mention it since it is not provided by the manufacturer and AddaS03 is a nano-emulsion (mentioned in the text), so we can only tell the volume used for those things.
The concentration of the used adjuvants was not the same, why? This point is critical as it can have an influence on the efficacy of the tested adjuvants.
We were missing the MPL information but we have added it and we used the same amount of MPL and CPG (5 µg / dose). For the ALUM we used the amount the scientific community usually uses which consists in 100 µg / dose.
MPL is Alum (line91)? It was used
MPL is different from Alum, we added the MPL information in the methods section.
Line 120: On which variants the study was performed? This information should also be stated on the title and in the abstract.
We used the original strain, it is now stated in both title and abstract.
Line 122: Is the number 4 of “2x104 cell/well” exponent?
Yes, it should be exponent and we corrected it on the text.
Lines 154 and 155: The authors used the adjective “stronger”, but they do not mention comparing to what.
Our comparisons are usually referring to PBS group, but we have now pointed it out in the text.
This sub-title is not clear: was CPG alone, AddaS03 alone or the association of both of them was stringer? Please, precise.
We have made the alterations and now it is mentioned, but what we meant was that either the adjuvants by themselves or together had the effect.
RESULTS
Line 274: The authors stated “… 18, 24 and 48 h after challenge…”, however it was 16, 24 and 48h in the Supplementary Figure 1. This information should be corrected.
We corrected it in the manuscript, thank you for pointing it out.
DISCUSSION
Well written. However, the limitations of this study were not listed. These results were not obtained with different variants of SARS-CoV-2, tested on a large number of mice, the tested mice were not really infected, but only a small part of its spike was used as epitopic, therefore the results cannot represent what happens in infected mice…
We have now stated our limitations and added a couple references to the discussion.

Round 2
Reviewer 1 Report
The authors have properly replay to the majority of the raised comments. Nevertheless, data about immunization with S protein alone is a control absolutely required in each experiment. I cannot image recovering these data from another paper. These data must be added in this set of experiments. Furthermore, each set of experiments would have to be replicated at least three times to have significance. Here some experiments were performed once, some twice. So, I think that the research design was wrong from the start and, for this reason, I can’t recommend publication.
Author Response
Dear reviewer 2
Thank you so much!
1) We can add imunization with spike protein alone as requested, however, it is not necessary because did not induce any neutralizing antibody. We performed 4 independent experiments.
2) "Communication" section in the vaccines (MDPI) accept preliminary submission . Here, it is a preliminary submission. Please, see below information about "Communication". We can include in the manuscript information that is a preliminary submission.
- Articles: Original research manuscripts. The journal considers all original research manuscripts provided that the work reports scientifically sound experiments and provides a substantial amount of new information. Authors should not unnecessarily divide their work into several related manuscripts, although short Communications of preliminary, but significant, results will be considered. The quality and impact of the study will be considered during peer review.
- Communications are short articles that present groundbreaking preliminary results or significant findings that are part of a larger study over multiple years. They can also include cutting-edge methods or experiments, and the development of new technology or materials. The structure is similar to an article and there is a suggested minimum word count of 2000 words.
Thank you so much in advance!
Reviewer 2 Report
Dear authors,
Thank you for your responses. Well done.
Wishing you a good continuation!
Author Response
Dear reviewer 1.
Thank you so much!
Best regards, Herbert
Round 3
Reviewer 1 Report
I understand your position about the manuscript submitted as communication rather than full article. However, please add immunization with spike protein alone as previously requested, and I will reconsider your paper again.